# A Proposal for a New Lung Ultrasound Score in Rheumatoid Arthritis: The Reliability of Lung Ultrasound for Rheumatoid Arthritis-Associated Interstitial Lung Disease Diagnosis

**DOI:** 10.3390/jcm14113701

**Published:** 2025-05-25

**Authors:** Esther Francisca Vicente-Rabaneda, Ingrid Möller, Abdon Mata, Nuria Montes, Gabriel-Santiago Rodríguez-Vargas, Luis Coronel, David Bong, Santos Castañeda, Pedro Santos-Moreno

**Affiliations:** 1Rheumatology Department, Hospital Universitario de La Princesa, IIS-Princesa, C/Diego de León 62, 28006 Madrid, Spain; nuria.montes.casado@gmail.com (N.M.); scastas@gmail.com (S.C.); 2Facultad de Medicina, Universidad Autónoma de Madrid, C/del Arzobispo Morcillo 4, 28029 Madrid, Spain; 3Instituto Poal de Reumatología, Carrer de Castanyer, 15, Sarrià-Sant Gervasi, 08022 Barcelona, Spain; ingrid.moller@ipoal.com (I.M.); dabong47@gmail.com (D.B.); 4Fundación Neumológica Colombiana, 13b Street #161-85, Bogotá 110221, Colombia; abdonmata@gmail.com; 5Research Department, Biomab IPS, 48th Street #13-86, Bogotá 110221, Colombia; santiago.rv18@gmail.com; 6Rheumatology Department, Barcelona Campus, Hospital Universitari Vall d’Hebron, 08035 Barcelona, Spain; luiscoroneltarancon@gmail.com; 7Bellvitge Campus, Universitat de Barcelona, Carrer de la Feixa Llarga s/n, L’Hospitalet de Llobregat, 08036 Barcelona, Spain; 8Rheumatology and Research Departments, Biomab IPS, 48th Street #13-86, Bogotá 110221, Colombia

**Keywords:** lung ultrasound, reliability, rheumatoid arthritis, interstitial lung disease, B lines, pleural line

## Abstract

**Background/Objectives**: The objective of this study is to investigate the intra- and inter-explorer reliability of different lung ultrasound (LUS) scores in patients with suspected rheumatoid arthritis with associated interstitial lung disease (RA-ILD). **Methods**: Fourteen consecutive patients with suspected RA-ILD based on the presence of respiratory symptoms, lung function tests (LFTs) or imaging alterations were recruited. The screening protocol consisted of an LFT, a chest X-ray, and HRCT. LUS examinations of different B-line and pleural line scores including 14 intercostal spaces were performed by three experienced sonographers, guaranteeing blinding. Intra- and inter-explorer reliability were calculated for all LUS scores and at the intercostal space level by weighted Cohen’s kappa and Fleiss’ kappa, respectively, relying on absolute differences using Stata/IC 14.2 software^®^ (StataCorp, College Station, TX, USA). **Results**: Both global (ĸ = 0.73–0.82) and binary (ĸ = 0.80–0.90) scores of B lines showed substantial to excellent intra-explorer reliability, with slightly better results for the binary score. The inter-explorer reliability was equally excellent for the global score (ĸ = 0.93) and the binary score (ĸ = 0.90) of B lines. The intra-explorer reliability of the semiquantitative pleural score was excellent for all the sonographers (ĸ = 0.88–0.91), and the binary pleural score had slightly lower intra-explorer reliability (ĸ = 0.77–0.84). Regarding inter-explorer reliability, both semiquantitative and binary pleural scores were equally excellent (ĸ = 0.84). Good to excellent inter-explorer reliability was found in all the scanned areas. **Conclusions**: Substantial to excellent intra- and inter-explorer reliability of different feasible B-line and pleural LUS scores were found, adding evidence in favor of the potential implementation of LUS for RA-ILD diagnosis in clinical practice.

## 1. Introduction

A clinical hallmark of rheumatoid arthritis (RA) is chronic symmetric and progressive polyarthritis, which can lead to structural damage and disability [1], but it can present extra-articular manifestations, as interstitial lung disease (ILD) is one of the most relevant diseases [2,3]. Clinically significant RA-associated ILD (RA-ILD) affects 5–10% of patients over their lifetime [4,5]. Its incidence has increased in recent years, motivated by greater concern about the disease and the availability of better diagnostic tools, mainly high-resolution chest computed tomography (HRCT), which is the current gold standard for its diagnosis [6].

The RA-ILD course is progressive in around half of the patients and is associated with shortened survival. Early diagnosis is crucial since treatment may potentially change its evolutionary course [7,8,9,10,11]. However, we still lack universally accepted guidelines on the diagnostic approach to RA-ILD. An expert-based proposal for the screening criteria has recently been published [12], focusing on an active search for risk factors [13,14] and respiratory symptoms and signs (dyspnea, cough and velcro crackles), lung function tests (LFTs) and radiological studies. Recently, the 2023 American College of Rheumatology (ACR)/American College of Chest Physicians (CHEST) guideline for the screening and monitoring of ILD in people with systemic autoimmune rheumatic diseases highlights the importance of the early detection of ILD, including subclinical diseases [15]. However, traditional exams such as LFTs and chest X-rays have limitations derived from the low sensitivity in the early stages of RA-ILD or, in the case of HRCT, related to costs, accessibility and ionizing radiation exposure [16].

In this context, additional diagnostic tools without the aforementioned drawbacks are needed, as lung ultrasound (LUS) is a promising candidate in light of its favorable sensitivity, specificity and negative predictive values, which could help better select patients for HRCT [17]. LUS has proven to be even more sensitive than chest auscultation, chest X-rays and LFTs in identifying the presence of ILD [18] and has the advantages of being non-invasive, having a relatively low cost and being accessible, providing immediate information during the consultation [19].

In line with the literature, a previous research study from our group carried out in 192 patients with RA and suspected RA-ILD also suggested the utility of LUS for ILD diagnosis since the presence of >11.5 B lines on the ROC curve predicted ILD presence (AUC 0.63; 95% CI: 0.55–0.71; *p* < 0.003) [20]. However, LUS has not yet completed its validation process. Data available on RA-ILD are still scarce and come from studies with a limited sample size and great heterogeneity in the characteristics of patients, examination protocols and equipment used, as well as definitions of pathology [17,21,22,23,24,25,26,27,28]. Precisely, the lack of consensus on the basic lesions to be evaluated and scores to be used, as well as the fact that LUS is an explorer-dependent technique, is among the main current limitation for its implementation in clinical practice for RA-ILD diagnosis.

Regarding the reliability of LUS for RA-ILD screening or diagnosis, the existing data are extremely limited and focus on the assessment of global B-line scores, generally, between two explorers or two observers of previously recorded images [22,23,29]. Therefore, the main objective of our study was to investigate the intra- and inter-explorer reliability of different scores of B-line and pleural line alterations in patients with RA and suspected RA-ILD. Our secondary objective was to compare the reproducibility of our LUS scores with that of previously published proposals about this topic obtained via a systematic review of the literature.

## 2. Materials and Methods

### 2.1. Patients

A total of fourteen consecutive patients with RA under investigation for suspected RA-ILD at a center specialized in the management of RA in Colombia were included in this reliability study conducted in August 2022. All patients met the 2010 ACR/European Alliance of Associations for Rheumatology (EULAR) classification criteria [30] and were undergoing routine clinical monitoring of their disease on an outpatient basis at the IPS Biomab. RA-ILD suspicion was based on the presence of velcro crackles or respiratory symptoms (chronic cough and/or dyspnea) in the last two months that could not be explained by a recent respiratory process (heart failure, infection, pleural effusion, atelectasis, asthma or chronic obstructive pulmonary disease), an LFT or imaging alterations. Pregnancy and previous interstitial lung involvement were additional exclusion criteria. The recruitment period was from May to July 2022. Eligible patients who agreed to participate in this study underwent analyses involving an LFT, HRCT and LUS.

The study was developed in accordance with the Declaration of Helsinki and approved by the local Institutional Research Ethics Committee at Hospital de San José, Bogota, Colombia (Record 0085–2021, 19 February 2021). All patients signed informed consent forms before participating in the procedures specified by the protocol.

### 2.2. Lung Ultrasound Investigation

Three physicians, two rheumatologists (EVR and IM) and one pulmonologist (AM) with extensive experience in ultrasound, including LUS for ILD screening, blindly and independently performed the ultrasound examination. To reinforce blinding between explorers, neither was present during the other’s examinations. Furthermore, to reinforce the blinding of the sonographers for the intra-explorer study, the examination room was darkened, and patients were instructed not to exchange comments with the sonographers. Sonographers were also blind to the patients’ diagnoses and clinical data. The time allocated for each complete LUS assessment was 15–20 min.

All the LUS examinations were performed during the same day in two rounds. In the first round, all patients (n = 14) were included in the inter-explorer reliability study, with each patient being scanned successively by the 3 sonographers, guaranteeing inter-explorer blinding. In the second round, 6 of those patients underwent an additional examination by all the sonographers, in the same conditions previously described, to estimate intra-explorer reliability. The overall examination time required for our intra- and inter-examiner reliability study was more than 5 h. Additionally, the two rounds of LUS examinations were separated by the necessary time interval to ensure rest for patients and examiners, as well as to avoid recall bias. Thus, a full day of work was required to carry out this reliability work.

The same model and initial settings of a real-time ultrasound machine, MyLab™Seven^®^ (Esaote, Genoa, Italy), were used by all the sonographers. Examinations were conducted with a multifrequency linear array probe (3–13 MHz). A single focus was set at the level of the pleural line. Depth was adjusted at 6–8 cm for the B-line evaluation and more superficially for the pleural line examination (4–5 cm) according to the patient’s phenotype. Automatic image-enhancement settings, including the second harmonic, were removed. Individualized adjustments directed to improve image quality according to the patient’s characteristics were allowed. An examination time of 15 min was set for each patient.

Pleural line and lung reverberation artifacts were evaluated in all patients to define either a healthy or pathological LUS pattern. A normal LUS pattern was defined by the presence of a normal pleural line and A lines. The pleural line was considered normal when a thin, regular hyperechoic line with its characteristic lung sliding was visualized. A lines were defined as horizontal reverberation artifacts with the appearance of replicas of the pleural line in depth at constant intervals [31]. The suspicion of ILD using LUS was defined by the presence of multiple B lines and/or abnormality of the pleural line [31,32,33,34,35]. B lines were defined as vertical reverberation artifacts appearing as discrete, laser-like vertical hyperechoic lines that arise from the pleural line and extend to the bottom of the screen without fading, erasing the A lines and moving synchronously with lung sliding. The pleural line was considered pathological when irregularity, thickening or fragmentation was visualized.

The abbreviated examination protocol of 14 intercostal spaces described by Gutiérrez et al. was performed due to its greater feasibility and the high sensitivity and specificity values previously published [25,36]. The 2nd intercostal space of the parasternal line; the 4th intercostal space of the mid-clavicular, anterior axillary and mid-axillary lines; and the 8th intercostal space of the paravertebral, subscapular and posterior axillary lines were scanned bilaterally. Patients were examined in the supine position, with the arms elevated above the head, for the evaluation of the anterolateral thorax, as well as in the sitting position, with the trunk slightly inclined forward and the arms at the sides, for the posterior thorax. Intercostal spaces were scanned longitudinally, following the anatomical lines previously described.

A global B-line score was calculated by adding the number of B lines of each intercostal space. In the case of the coalescence of B lines, their numbers were calculated as the percentage of the extension of the pleural line that they occupied divided by 10 [37]. When the number of B lines was ≥6, a value of 6 was arbitrarily assigned by agreement, as in our experience, the reproducibility of the B-line count is optimal up to a maximum of 5 B lines per intercostal space.

Additionally, we also counted B lines in a binary way: the presence of <3 B lines in an intercostal space was considered normal and assigned a value of 0, while ≥3 B lines defined the intercostal space as pathological and were counted as 1, following the definition of a positive region established by the International Evidence-Based Recommendations for Point-of-Care Lung Ultrasound, which is agreed upon by experts and widely used in other medical fields such as the assessment of interstitial syndrome in emergencies and in critically ill patients [38]. Accordingly, a binary B-line score was generated by adding the binary values of B lines in all the intercostal spaces.

Pleural line measurements were carried out in two ways. The presence of abnormalities was quantified as 1 and their absence as 0 to generate a binary score by adding the values of all intercostal spaces scanned (range: 0–14). Additionally, a semiquantitative pleural score (range: 0–42) assessed both the presence and extension of alterations in the pleural line: “0” for normal (no alterations in the pleural line), “1” for mild alterations affecting <30% of the width of the pleural line, “2” for moderate alterations of 30–50% of its width and “3” marked alterations and/or extension of >50% of the pleural line.

A prior patient-based consensus was reached on the interpretation of LUS findings by means of a brief evaluation of 2 intercostal spaces from the anterior, lateral and posterior thorax carried out by the most experienced sonographer (IM) on a patient that was not included in the analysis.

### 2.3. Systematic Review of the Literature

The objective of the systematic review was to gather the existing evidence in the literature about the reliability of LUS applied to the screening or diagnosis of RA-ILD. We searched for original papers about the use of LUS for RA-ILD screening or diagnosis published in PUBMED, EMBASE or the Cochrane Library until the end of December 2024 in the Title/Abstract or Title Abstract Keyword sections, respectively, using the following search strategies according to each database: (a) PUBMED: (“rheumatoid arthritis” [Title/Abstract] AND (“interstitial lung disease” [Title/Abstract] OR “ILD” [Title/Abstract] OR “usual interstitial pneumo*”[Title/Abstract]) AND (“ultrasound” [Title/Abstract] OR “ultrasonography*” [Title/Abstract])) AND (1000/1/1:2024/12/31[pdat]); (b) EMBASE: ‘rheumatoid arthritis’:ab,ti AND (‘interstitial lung disease’:ab,ti OR ‘ild’:ab,ti OR ‘usual interstitial pneumonia’:ab,ti) AND (‘ultrasound’:ab,ti OR ‘ultrasonography’:ab,ti) AND (‘article’/it OR ‘article in press’/it OR ‘clinical trial’/it OR ‘erratum’/it OR ‘review’/it) AND [<1966–2024]/py, excluding conference abstracts and conference reviews; and (c) Cochrane Library: “rheumatoid arthritis” AND (“interstitial lung disease” OR “ILD” OR (usual interstitial NEXT pneumo*)) AND (“ultrasound” OR “ultrasonography”).

The article selection process began with a review of the title and abstract. If information on the reliability of LUS was not included, the materials and methods sections and finally the results were reviewed. Only original manuscripts in which the reliability of LUS for RA-ILD screening or diagnosis was measured and reported were selected, excluding reviews, meta-analyses, editorials and case reports. Despite the reviews not being eligible for inclusion, they were revised for potential references of interest that met the inclusion criteria and had not been identified by the search strategy. The following data were extracted from the selected articles: author, year of publication, patients included, type of ultrasound machines used, scanning protocols and scores and reliability data.

### 2.4. Statistical Analysis

Intra- and inter-explorer reliability were calculated for all LUS scores using weighted Cohen’s kappa and Fleiss’ kappa, respectively, relying on absolute differences. The 95% confidence interval (CI) of kappa and the degree of agreement were also evaluated. Reliability was also estimated at the intercostal space level. Kappa values were interpreted according to Landis et al. [39]. The ability of our LUS scores to discriminate the presence of ILD in HCRT was assessed using receiver operating characteristic (ROC) curves, and optimal cut-off values were determined. The sensitivity, specificity, and area under the ROC curve (AUROC) were considered as diagnostic performance indicators. Statistical analysis was performed using the Stata/IC 14.2 software^®^ (StataCorp LLC, College Station, TX, USA).

## 3. Results

The 14 intercostal spaces of all patients were explored, both for the presence of B lines and for the identification of alterations in the pleural line. Taking into account the examination time per patient and explorer in two rounds (>5 h), the time devoted to the previous patient-based consensus of the interpretation of LUS findings and the rest time needed for the evaluators and patients, as well as the recommended time to avoid the bias of having the slightest memory of what had been previously assessed, our reliability study took a full day of work. The main clinical characteristics of the population included in this study are displayed in Appendix A.

### 3.1. Reliability of the B-Line Scores

The intra-explorer reliability of the different scores of B lines measured as Kappa (95% CI) and the observed agreement are displayed in Table 1. Both the global (ĸ = 0.73–0.82) and the binary (ĸ = 0.80–0.90) scores of B lines showed substantial to excellent intra-explorer reliability, with slightly better results for the binary score. Regarding the binary B-line score, the three explorers had a kappa of one in more than 50% of the intercostal spaces scanned (Table 1).

The inter-explorer reliability was equally excellent for the global (ĸ = 0.93) and the binary (ĸ = 0.90) scores of B lines, with all the intercostal spaces displaying good to excellent inter-explorer reliability (Table 2).

### 3.2. Reliability of the Pleural Line Scores

The intra-explorer and inter-explorer kappa (95% CI) and observed agreement for the pleural line scores are shown in Table 3 and Table 4, respectively.

The intra-explorer reliability of the semiquantitative pleural score was excellent for all the sonographers (ĸ = 0.88–0.91), although the variability found between the examiners at the level of the intercostal spaces was greater than that for B-line scores. The only locations with a moderate kappa result for at least one of the sonographers were the second right parasternal line, the second left parasternal line, and the fourth left anterior axillary line. The binary pleural score had slightly lower intra-explorer reliability (ĸ = 0.77–0.84), with more locations with at least one moderate kappa measure (Table 3).

Regarding inter-explorer reliability, both semiquantitative and binary pleural scores were equally excellent (ĸ = 0.84), with good to excellent results in all the scanned areas (Table 4).

### 3.3. Diagnostic Performance of the Proposed LUS Scores

Although the objective of our research focused on reliability and not the accuracy of LUS, we evaluated whether our proposed LUS scores could discriminate against the presence of ILD on HRCT, which is considered the gold standard for the diagnosis of RA-ILD. HRCT was normal in four patients, showing ILD in the rest. The values of the AUROC were generally good for all scores and examiners, ranging from 0.74 to 0.87. Table 5 shows the optimal cut-off points for each LUS score, which were obtained using ROC curves, to discriminate between patients with and without ILD on the HRCT, AUROC, sensitivity and specificity.

Examples of LUS examinations with different degrees of B-line and pleural line alterations are shown in Figure 1.

### 3.4. Systematic Literature Review

Forty manuscripts were identified using the search strategy in the PUBMED and EMBASE databases and two manuscripts in the Cochrane Library. Six eligible papers were selected after excluding those that did not meet the inclusion criteria and duplicates (Figure 2). No additional original articles were identified after revising the reviews.

The main characteristics of the six selected manuscripts are shown in Table 6 [22,23,29,40,41,42].

Overall, there was great heterogeneity in terms of included patients, scanning protocols, ultrasound machines, types of probes, definitions of the elementary lesions and proposed scores. None of them described intra-explorer reliability, and only two studies evaluated the intra-observer reliability of one and two raters, respectively [40,41]. Inter-explorer reliability was only investigated by Cogliati et al. between two examiners [22], while a mention about inter-observer reliability was found in all the selected studies and was related either to the different number of sonographers or raters [22,23,29,40,41,42]. Information about the reliability exercises was scarce in most of the papers. Only Gutiérrez et al. [29] provided more detailed information on how they conducted their inter-observer evaluation that was performed prior to the study to reach a consensus between sonographers, and they included eight patients with different connective tissue diseases associated with ILD (CTD-ILD) per examiner.

## 4. Discussion

The main objective of our study was to assess the reliability of LUS in patients with suspected RA-ILD using different approaches in terms of elementary lesions and scores. We found that either B-line or pleural line evaluations yielded good to excellent intra- and inter-explorer reliability: a kappa score of 0.84 for both pleural line scores and a kappa score of 0.90 or 0.93 for the semiquantitative or binary scores of B lines, respectively. One of the main aspects of our work to highlight is the extensive reliability study carried out, which evaluated both the intra- and inter-explorer reliability of two types of LUS scores for each elementary lesion, and we even assessed all these aspects at the intercostal space level.

In relation to the assessment of B lines, we found that the binary score, which was chosen according to some expert recommendations [38,43], yielded similar intra- and inter-explorer reliability values to the global score, which has been used more frequently in previous studies of CTD-ILD, especially scleroderma (SSc-ILD) [44,45,46,47] and RA-ILD [20,22,23,26,27,28,29]. We consider this finding of great interest since it can simplify the interpretation and quantification of LUS results, increasing its feasibility.

Furthermore, we believe that the excellent results of the intra- and inter-explorer reliability of pleural line scores are particularly noteworthy, since pleural alterations have been studied very little to date, not only in RA-ILD but also in SSc-ILD [23,40,41,48,49,50,51,52,53,54,55]. We want to highlight the good performance of our semiquantitative pleural score that considers not only the severity but also the extent of the affectation, providing more detailed information than the binary pleural score. Our training as rheumatologists to quantify synovitis semi-quantitatively using ultrasound could be one of the explanations for the good results of the semiquantitative pleural score, although the results of the pulmonologist sonographer were equally good. The superficial location of the pleural line, which facilitates its exploration, as it is less interfered with by the patient’s characteristics, may also have influenced the results. Interestingly, a pleural line examination is faster and simpler than investigating the presence of B lines and can be performed very adequately using linear probes, which are the most frequently available in rheumatology consultations. The fact that our semiquantitative pleural score does not require measurements makes it even simpler and more feasible, unlike other proposals that define the pathology of the pleural line by a thickness above a threshold that varies depending on the use of linear or convex probes, respectively [23,48,49,53,56]. In fact, none of our examinations lasted more than 15 min, despite assessing both the presence of B lines and pleural line pathology using various scores, which supports the feasibility of LUS for ILD screening.

At the intercostal space level, intra- and inter-explorer reliability were generally very good, although in some areas, there seemed to be less agreement. In the case of B lines, the binary score yielded slightly better results than the global score. Furthermore, some left intercostal spaces of the anterolateral thorax, especially when using the global B-line score, showed slightly lower reliability values, probably due to the difficulties generated by the presence of the heart. By contrast, semiquantitative pleural score results were slightly better than those of the binary pleural score, and the differences between the right and left anterolateral thorax were insignificant, which reinforces the interest of pleural evaluations. We found no substantial changes between the anterior and posterior thorax evaluations of the presence of both B lines and pleural line alterations.

To give robustness to our study and minimize additional sources of variability, we implemented all measures within our reach. First, we defined the elementary lesions to be evaluated (B lines and pleural line alterations), decided the scanning protocol and generated the corresponding scores, relying on the international evidence-based recommendations for point-of-care LUS [38] and previous data on LUS for SSc-ILD [18,44,45,46,53,57,58,59,60,61] and CTD-ILD [17,22,23,26,27,28,29,48,49,52,62,63,64,65], as well as our personal experience.

Despite most LUS studies focused on B lines, we decided to also include the evaluation of the pleural line due to its greater discriminative capacity between pathology and normality suggested in preliminary studies [23,47]. Moazedi-Fuerst et al. compared the frequency of LUS abnormalities in patients with RA without the suspicion of ILD and healthy controls [23]. They evaluated the presence of B lines and a fragmented pleural line in both groups. Interestingly, while B lines on at least two of the examined locations were observed in 7% of the healthy controls, none of them showed fragmentation of the pleural line, suggesting the importance of considering both alterations.

We chose the lung scanning protocol for 14 intercostal spaces due to its good balance and feasibility, sufficiently comprehensive examination, good sensitivity and specificity and negative predictive values described for both SSc-ILD and RA-ILD [29,36]. In fact, in the meta-analysis by Xie et al. [25], in which the diagnostic accuracy of LUS scores using different intercostal space counts was compared, the authors found that the score for the 14 intercostal spaces had better diagnostic performance than the global count that included 72 intercostal spaces.

Regarding LUS scores, our selection also reflects a combination of previous evidence from the literature, and our experience in musculoskeletal ultrasound as semiquantitative scores has demonstrated better reliability and informative capacity than binary scores. Finally, we did a practical patient-based exercise to further improve the consensus about the definitions of the elementary lesions, the scanning protocol and the interpretation of the LUS findings.

Technical issues are another key point to control when studying reliability. For that purpose, all the sonographers used the same ultrasound machine, probe, settings, and examination room in successive rotating shifts for each patient, guaranteeing blinding among explorers. Furthermore, a resting interval between the two rounds of LUS examinations was established to prevent recall bias for intra-explorer reliability evaluations.

Several reasons led us to choose the linear probe for both B-line and pleural line evaluations. First, it was used to bring our study closer in determining the reality of rheumatology in clinical practice, in which multifrequency linear probes are the ones available. Second, it was used to speed up and make examinations more feasible. Lastly, although key to this decision, it was chosen due to its good correlation and comparable precision reported between different types of probes [21,46]. Delle Sedie et al. compared the performance of cardiac and linear probes to identify B lines and their correlation with HRCT diagnosis of ILD in patients with systemic sclerosis [46]. They found a good correlation between them in terms of the evaluation of B lines, with an intraclass correlation coefficient (ICC) of 0.681. with regard to the correlation with HRCT, both probes displayed good sensitivity (85%), with a slight difference in specificity. Furthermore, linear probes allow for the better characterization of the pleural line, which was a key objective of our study, as recognized by some expert recommendations [43].

Regarding the literature, none of the manuscripts published to date on LUS in RA-ILD provide intra-explorer reliability data, while inter-explorer reliability has only been reported by one study (Table 5). Cogliati et al. found a kappa of 0.78 for the presence of ILD by comparing B-line scores obtained via an exam of 72 intercostal spaces performed with a standard ultrasound machine by an experienced examiner and that performed using a pocket-sized device by a more novice explorer, both with a convex probe [22]. Our inter-explorer reliability values for both B-line scores are higher than those described by Cogliati et al. (ĸ = 0.90 and 0.93 versus ĸ = 0.78, respectively), although the extensive experience of our sonographers could have played a role [22].

It should be noted that there is also scarce and not very detailed data on inter-observer reliability evaluated from stored images or videos, and some reliability data come from exams conducted prior to the study to reach a consensus. Only Gutierrez et al. [29] described with more detail their reliability exercise that included eight patients with CTD-ILD, which was independently assessed by the two examiners by counting B lines in 14 intercostal spaces with a multifrequency linear probe. Their reported inter-observer agreement (ĸ = 0.82) was excellent, although their value was slightly lower than ours (ĸ = 0.93). Furthermore, Cogliati et al. described that their previously reported r-value for inter-observer variability for the quantification of B lines was 0.96 [22,66], and Watanabe et al. [42] focused on inter-rater reliability for the total B-line measurements.

Other authors have conducted inter-observer reliability studies of pleural line alterations. Moazedi-Fuerst et al. [23] used the LUS score including 18 intercostal spaces that combined the evaluation of B lines (convex probe) and pleural line alterations (linear probe). They described an inter-observer kappa of 0.92 for ILD presence between the two explorers but did not assess the reliability by elementary lesion or region examined, making it more difficult to establish comparisons with our study. More recently, Bandinelli et al. have published the intra- and inter-reader ICC of semiquantitative pleural and parenchymal ultrasound scores, which range from 0 to 3 according to the following descriptions: a) pleural score: 0 = normal, 1 = thickened pleural line, 2 = fragmented pleural line and 3 = subpleural consolidation and b) parenchymal score: 0 = absent B lines, 1 = discrete and divergent B lines, 2 = confluent B lines and 3 = whiteout [40]. Examinations of fourteen intercostal spaces were performed using a linear probe by an expert senior sonographer. Saved images were additionally evaluated by three junior residents that had been trained over a 6-month period. They described intra-reader ICC > 0.9 for both scores (senior examiner) and good to excellent inter-reader ICC (0.82–0.84 for pleural alterations and 0.86–0.94 for parenchymal score) for the residents, in line with our findings. Finally, Zabaleta et al. [41] have also described inter-observer and intra-observer reliability data for B lines and pleural irregularities, which were obtained prior to their research study on LUS.

Despite a growing interest in evaluating the usefulness of pleural line abnormalities, RA-ILD screening has emerged in recent years, and reliability exercises are frequently lacking [56,67]. Vermant et al. evaluated the number and anatomic distribution of B lines, as well as the frequency of pleural line alterations, which are defined by the existence of thickening or fragmentation, and the presence of subpleural nodules or pleural effusions [56]. However, they did not describe the anatomic distribution of pleural line alterations. Interestingly, these authors described inter-rater reliability data from a previous study of LUS in patients with SARS-CoV-2 using the 12-zone protocol and a convex probe: kappa values of 0.79 for normal LUS and the presence of B lines, 0.16 for the total count of B lines, 0.23 for pleural thickening and 0.49 for subpleural consolidation and pleural effusion [68]. These data reinforce the convenience of using linear probes for the study of the pleural line.

Sofíudóttir et al. have proposed a new definition of pathologic LUS that considers both B lines and pleural line abnormalities [67]. They defined LUS as positive if the number of B lines was ≥10 in at least two of the examined areas (14 intercostal spaces) or the pleural line was thickened and fragmented in at least one area. Despite planning to study reliability in their published protocol [69], no reliability data were reported [67].

Additionally, Fairchild et al. proposed focusing ultrasound screening for ILD on the exclusive evaluation of the pleural line using a linear probe and a reduced count of 14 intercostal spaces [55]. They suggested assessing the presence of discontinuity or cavitation of the pleural line by considering the lung ultrasound examination to be positive if these abnormalities are present in at least one intercostal space, are larger than 2 mm in size, show pseudo-thickening and move with pleural sliding. A preliminary validation approach in patients with systemic sclerosis showed adequate sensitivity and specificity (100% and 82%, respectively), reporting an inter-observer reliability exercise with a kappa value of 0.82, although they have not conducted intra- or inter-explorer reliability exercises, nor have they assessed the performance of this pleural score in patients with RA. Based on their results, these authors proposed using LUS as a tool for selecting patients to perform HRCT.

The progressive incorporation of the evaluation of pleural line abnormalities in the most recent works denotes the interest it arouses, which gives added value to our intra- and inter-explorer reliability studies that incorporate the evaluation of the reliability of both B lines and pleural line alterations while considering different alternatives, such as binary and semiquantitative scores. We would like to emphasize that our proposed semiquantitative pleural score considers not only the presence of pleural line abnormalities but also their extent as a potential measure of severity.

In summary, demonstrating the reliability of LUS is of crucial importance to facilitate its implementation in clinical practice for the study of CTD-ILD such as RA-ILD. Despite operator dependency being one of the main disadvantages traditionally attributed to ultrasound, our good to excellent intra- and inter-explorer reliability results reinforce the usefulness of LUS for RA-ILD screening and represent a step forward in its validation process, highlighting the importance of examining pleural line alterations. We have not found a study of similar characteristics published to date and believe that our work helps to shed light on an unmet need, thereby adding relevant scientific evidence to the literature. As we have highlighted in the Discussion Section, no such comprehensive studies have been published to date on the inter- and intra-explorer reliability of lung ultrasound.

Based on our results, we propose that RA-ILD screening using LUS should include the evaluation of both B lines and pleural line alterations, and we believe that linear probes are adequate for both purposes. Additionally, the scores we propose seem more reproducible than those previously suggested and could simplify the interpretation and quantification of LUS results, which might improve their feasibility and implementation in clinical practice. In this regard, we highlight the good performance of our semiquantitative pleural score that considers the presence and severity of the affectation.

As a future perspective, given the favorable findings of our pilot study, we believe it would be of great interest to conduct further research to validate these results in studies with larger sample sizes by comparing LUS reliability between different levels of severity of RA-ILD, including patients in the preclinical or early stages of the disease. It would also be interesting to compare reliability between patients with RA-ILD and other systemic autoimmune rheumatic disease-associated ILDs.

### Limitations

The main limitations of our study are its limited sample size, monocentric nature and the successive inclusion of patients, without selection for different degrees of lung involvement. In our pilot reliability study, we applied measures at our disposal to mitigate the limitations of the small sample size, such as being conducted by experienced sonographers, as well as the previously mentioned measures used to increase this study’s robustness. Organizing reliability studies like ours, in which patients are examined twice by three explorers, is complex. They need to be conducted at a single site with a limited sample size and sufficient time (>5 h) to ensure blinding. However, the findings of our pilot study must be interpreted with caution and considered a starting point worthy of further investigation because its small sample size limits the generalizability of the conclusions.

Due to the recruitment of patients being consecutive and without randomization, the existence of a selection bias cannot be ruled out. Additionally, it did not allow for stratification by RA-ILD severity, although its impact on reliability might not be possible to delineate with the limited sample size. Finally, the inherent limitations of LUS in its ability to detect pulmonary fibrotic patterns compared to HRCT cannot be completely ruled out. B lines may be present in other pathologies such as heart failure, pulmonary infections, pleural disease or atelectasis, which limits its specificity, so a thorough clinical evaluation is necessary to exclude these processes. Furthermore, in healthy elderly patients, a limited number of B lines can also be found, which can complicate the interpretation of LUS in patients with suspected RA-ILD due to its association with age as one of its risk factors. It should be added that LUS, unlike HRCT, cannot evaluate the entire parenchyma and has limitations for differential diagnosis with infection and cancer.

## 5. Conclusions

Our study demonstrates the substantial to excellent intra- and inter-explorer reliability of simple and feasible LUS scores, which evaluate B lines and pleural line alterations, and seems more reproducible than those previously proposed. Our results add evidence in favor of the potential implementation of LUS in clinical practice as an aid tool for the diagnosis of RA-ILD and more efficiently select patients who are candidates for HRCT, deserving further validation in studies with larger sample sizes and stratification by RA-ILD severity.

## Figures and Tables

**Figure 1 jcm-14-03701-f001:**
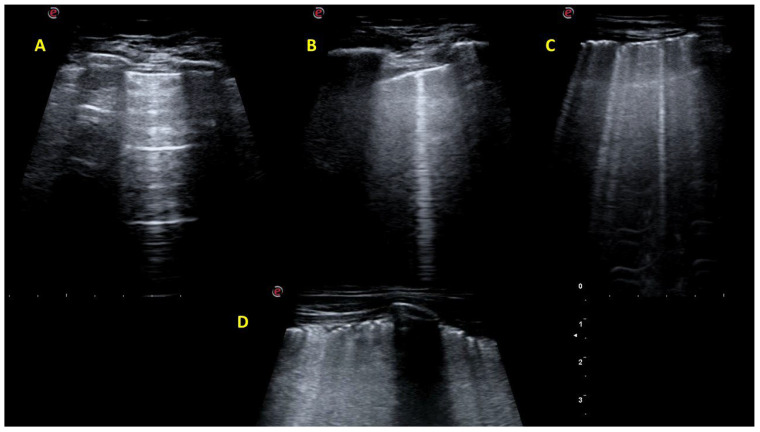
Lung ultrasound examinations of the patients included in the study. Examples of lung ultrasound (LUS) examinations with different degrees of affectation. (**A**) Normal LUS pattern. (**B**) Single B line. (**C**) Multiple B lines with mild pleural line alterations. (**D**) Coalescent B lines with marked pleural line alteration.

**Figure 2 jcm-14-03701-f002:**
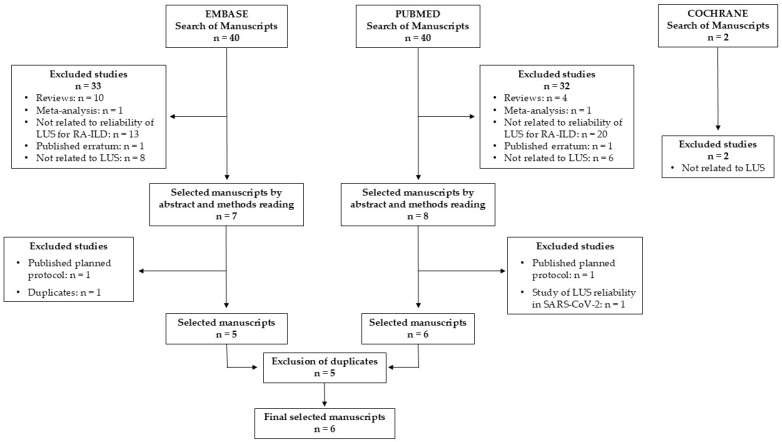
Systematic review flow chart. LUS: lung ultrasound. RA-ILD: rheumatoid arthritis with associated interstitial lung disease.

**Table 1 jcm-14-03701-t001:** Intra-explorer reliability of the B-line scores.

B-Line Evaluations	B-Line Global Count		B-Line Binary Count	
	Kappa (95%CI)	Agreement %	Kappa (95%CI)	Agreement %
Intercostal Spaces				
Anterolateral Thorax				
2nd right parasternal line	E1: 0.86 (0.49–1.00)	E1: 94.4	E1: 1.00 (1.00–1.00)	E1: 100
	E2: 0.59 (−0.03–1.00)	E2: 83.3	E2: 1.00 (1.00–1.00)	E2: 100
	E3: 1.00 (1.00–1.00)	E3: 100	E3: 1.00 (1.00–1.00)	E3: 100
2nd left parasternal line	E1: 0.61 (−0.13–1.00)	E1: 83.3	E1: 1.00 (1.00–1.00)	E1: 100
	E2: 0.87 (0.52–1.00)	E2: 94.4	E2: 1.00 (1.00–1.00)	E2: 100
	E3: 0.40 (−0.33–1.00)	E3: 72.2	E3: 1.00 (1.00–1.00)	E3: 100
4th right mid-clavicular line	E1: 1.00 (1.00–1.00)	E1: 100	E1: 1.00 (1.00–1.00)	E1: 100
	E2: 0.75 (0.04–1.00)	E2: 91.7	E2: 1.00 (1.00–1.00)	E2: 100
	E3: 1.00 (1.00–1.00)	E3: 100	E3: 1.00 (1.00–1.00)	E3: 100
4th left mid-clavicular line	E1: 1.00 (1.00–1.00)	E1: 100	E1: 1.00 (1.00–1.00)	E1: 100
	E2: 1.00 (1.00–1.00)	E2: 100	E2: 1.00 (1.00–1.00)	E2: 100
	E3: 0.75 (0.04–1.00)	E3: 91.7	E3: 1.00 (1.00–1.00)	E3: 100
4th right anterior axillary line	E1: 1.00 (1.00–1.00)	E1: 100	E1: 1.00 (1.00–1.00)	E1: 100
	E2: 1.00 (1.00–1.00)	E2: 100	E2: 1.00 (1.00–1.00)	E2: 100
	E3: 1.00 (1.00–1.00)	E3: 100	E3: 1.00 (1.00–1.00)	E3: 100
4th left anterior axillary line	E1: 1.00 (1.00–1.00)	E1: 100	E1: 1.00 (1.00–1.00)	E1: 100
	E2: 0.60 (−0.15–1.00)	E2: 83.3	E2: 0.57 (−0.42–1.00)	E2: 83.3
	E3: 0.73 (0.23–1.00)	E3: 88.9	E3: 0.25 (−0.91–1.00)	E3: 66.7
4th right mid-axillary line	E1: 0.86 (0.49–1.00)	E1: 94.4	E1: 0.86 (0.49–1.00)	E1: 94.4
	E2: 1.00 (1.00–1.00)	E2: 100	E2: 1.00 (1.00–1.00)	E2: 100
	E3: 0.86 (0.48–1.00)	E3: 94.4	E3: 1.00 (1.00–1.00)	E3: 100
4th left mid-axillary line	E1: 0.61 (−0.08–1.00)	E1: 83.3	E1: 0.33 (−0.47–1.00)	E1: 66.7
	E2: 0.85 (0.47–1.00)	E2: 94.4	E2: 0.57 (−0.42–1.00)	E2: 83.3
	E3: 0.78 (0.20–1.00)	E3: 91.7	E3: 0.67 (−0.14–1.00)	E3: 83.3
**Posterior Thorax**				
8th right paravertebral line	E1: 0.86 (0.49–1.00)	E1: 94.4	E1: 1.00 (1.00–1.00)	E1: 100
	E2: 1.00 (1.00–1.00)	E2: 100	E2: 1.00 (1.00–1.00)	E2: 100
	E3: 0.85 (0.47–1.00)	E3: 94.4	E3: 1.00 (1.00–1.00)	E3: 100
8th left paravertebral line	E1: 0.75 (0.04–1.00)	E1: 91.2	E1: 1.00 (1.00–1.00)	E1: 100
	E2: 0.85 (0.47–1.00)	E2: 94.4	E2: 1.00 (1.00–1.00)	E2: 100
	E3: 0.78 (0.20–1.00)	E3: 91.7	E3: 1.00 (1.00–1.00)	E3: 100
8th right subscapular line	E1: 0.71 (0.18–1.00)	E1: 88.9	E1: 0.57 (−0.42–1.00)	E1: 83.3
	E2: 0.86 (0.47–1.00)	E2: 94.4	E2: 1.00 (1.00–1.00)	E2: 100
	E3: 0.78 (0.20–1.00)	E3: 91.7	E3: 1.00 (1.00–1.00)	E3: 100
8th left subscapular line	E1: 1.00 (1.00–1.00)	E1: 100	E1: 1.00 (1.00–1.00)	E1: 100
	E2: 0.85 (0.47–1.00)	E2: 94.4	E2: 0.57 (−0.42–1.00)	E2: 83.3
	E3: 0.74 (0.07–1.00)	E3: 91.7	E3: 0.67 (−0.14–1.00)	E3: 83.3
8th right posterior axillary line	E1: 0.71 (−0.03–1.00)	E1: 88.9	E1: 1.00 (1.00–1.00)	E1: 100
	E2: 0.86 (0.48–1.00)	E2: 94.4	E2: 1.00 (1.00–1.00)	E2: 100
	E3: 0.86 (0.49–1.00)	E3: 94.4	E3: 1.00 (1.00–1.00)	E3: 100
8th left posterior axillary line	E1: 1.00 (1.00–1.00)	E1: 100	E1: 1.00 (1.00–1.00)	E1: 100
	E2: 1.00 (1.00–1.00)	E2: 100	E2: 1.00 (1.00–1.00)	E2: 100
	E3: 1.00 (1.00–1.00)	E3: 100	E3: 1.00 (1.00–1.00)	E3: 100
**Global and Binary Scores of B Lines, Respectively**	E1: 0.73 (0.39–1.00)	E1: 88.9	E1: 0.90 (0.39–1.00)	E1: 95.8
	E2: 0.80 (0.55–1.00)	E2: 91.7	E2: 0.84 (0.56–1.00)	E2: 93.3
	E3: 0.82 (0.56–1.00)	E3: 92.9	E3: 0.80 (0.45–1.00)	E3: 91.7

CI: confidence interval; E1: explorer 1; E2: explorer 2; E3: explorer 3. The intercostal spaces with the best intra-explorer kappa results are colored in green or orange for the global score or the binary B-line score, respectively.

**Table 2 jcm-14-03701-t002:** Inter-explorer reliability of the B-line scores.

B-Line Evaluations	B-Lines Global Score		B-Line Binary Score	
	Kappa (95%CI)	Agreement %	Kappa (95%CI)	Agreement %
**Intercostal Spaces**				
**Anterolateral Thorax**				
2nd right parasternal line	0.84 (0.62–1.00)	93.7	1.00 (1.00–1.00)	100
2nd left parasternal line	0.75 (0.45–1.00)	90.5	1.00 (1.00–1.00)	100
4th right mid-clavicular line	0.95 (0.83–1.00)	98.4	1.00 (1.00–1.00)	100
4th left mid-clavicular line	1.00 (1.00–1.00)	100	1.00 (1.00–1.00)	100
4th right anterior axillary line	0.96 (0.85–1.00)	98.4	1.00 (1.00–1.00)	100
4th left anterior axillary line	0.79 (0.46–1.00)	93.7	0.69 (0.21–1.00)	90.5
4th right mid-axillary line	1.00 (1.00–1.00)	100	1.00 (1.00–1.00)	100
4th left mid-axillary line	0.93 (0.77–1.00)	97.6	0.83 (0.44–1.00)	95.2
**Posterior Thorax**				
8th right paravertebral line	0.85 (0.65–1.00)	94.0	0.88 (0.62–1.00)	95.2
8th left paravertebral line	0.97 (0.89–1.00)	98.8	1.00 (1.00–1.00)	100
8th right subscapular line	0.91 (0.75–1.00)	96.4	0.84 (0.53–1.00)	95.2
8th left subscapular line	0.69 (0.37–1.00)	88.1	0.61 (0.16–1.00)	85.7
8th right posterior axillary line	0.81 (0.55–1.00)	92.9	0.87 (0.58–1.00)	95.2
8th left posterior axillary line	1.00 (1.00–1.00)	100	1.00 (1.00–1.00)	100
**Global and Binary Scores of B Lines, Respectively**	0.93 (0.86–1.00)	97.1	0.90 (0.76–1.00)	96.0

CI: confidence interval.

**Table 3 jcm-14-03701-t003:** Intra-explorer reliability of the pleural line scores.

Pleural Line Evaluations	Semiquantitative Score		Binary Score	
	Kappa (95%CI)	Agreement %	Kappa (95%CI)	Agreement %
Intercostal Spaces				
Anterolateral Thorax				
2nd right parasternal line	E1: 1.00 (1.00–1.00)	E1: 100	E1: 1.00 (1.00–1.00)	E1: 100
	E2: 0.54 (−0.42–1.00)	E2: 83.3	E2: 0.25 (−0.91–1.00)	E2: 66.7
	E3: 1.00 (1.00–1.00)	E3: 100	E3: 1.00 (1.00–1.00)	E3: 100
2nd left parasternal line	E1: 1.00 (1.00–1.00)	E1: 100	E1: 1.00 (1.00–1.00)	E1: 100
	E2: 1.00 (1.00–1.00)	E2: 100	E2: 1.00 (1.00–1.00)	E2: 100
	E3: 0.57 (−0.21–1.00)	E3: 83.3	E3: 0.33 (−0.47–1.00)	E3: 66.7
4th right mid-clavicular line	E1: 1.00 (1.00–1.00)	E1: 100	E1: 1.00 (1.00–1.00)	E1: 100
	E2: 0.75 (0.04–1.00)	E2: 91.7	E2: 0.57 (−0.42–1.00)	E2: 83.3
	E3: 1.00 (1.00–1.00)	E3: 100	E3: 1.00 (1.00–1.00)	E3: 100
4th left mid-clavicular line	E1: 1.00 (1.00–1.00)	E1: 100	E1: 1.00 (1.00–1.00)	E1: 100
	E2: 1.00 (1.00–1.00)	E2: 100	E2: 1.00 (1.00–1.00)	E2: 100
	E3: 1.00 (1.00–1.00)	E3: 100	E3: 1.00 (1.00–1.00)	E3: 100
4th right anterior axillary line	E1: 1.00 (1.00–1.00)	E1: 100	E1: 1.00 (1.00–1.00)	E1: 100
	E2: 1.00 (1.00–1.00)	E2: 100	E2: 1.00 (1.00–1.00)	E2: 100
	E3: 0.67 (−0.14–1.00)	E3: 83.3	E3: 0.67 (−0.14–1.00)	E3: 83.3
4th left anterior axillary line	E1: 0.75 (0.04–1.00)	E1: 91.7	E1: 0.57 (−0.42–1.00)	E1: 83.3
	E2: 0.57 (−0.21–1.00)	E2: 83.3	E2: 0.33 (−0.47–1.00)	E2: 66.7
	E3: 0.57 (−0.42–1.00)	E3: 83.3	E3: 0.57 (−0.42–1.00)	E3: 83.3
4th right mid-axillary line	E1: 1.00 (1.00–1.00)	E1: 100	E1: 1.00 (1.00–1.00)	E1: 100
	E2: 0.78 (0.16–1.00)	E2: 91.7	E2: 0.67 (−0.14–1.00)	E2: 83.3
	E3: 1.00 (1.00–1.00)	E3: 100	E3: 1.00 (1.00–1.00)	E3: 100
4th left mid-axillary line	E1: 0.78 (0.70–1.00)	E1: 91.7	E1: 0.67 (−0.14–1.00)	E1: 83.3
	E2: 0.75 (0.04–1.00)	E2: 91.7	E2: 0.57 (−0.42–1.00)	E2: 83.3
	E3: 1.00 (1.00–1.00)	E3: 100	E3: 1.00 (1.00–1.00)	E3: 100
**Posterior Thorax**				
8th right paravertebral line	E1: 0.78 (0.16–1.00)	E1: 91.7	E1: 0.67 (−0.14–1.00)	E1: 83.3
	E2: 0.78 (0.20–1.00)	E2: 91.7	E2: 0.67 (−0.14–1.00)	E2: 83.3
	E3: 0.75 (0.04–1.00)	E3: 91.7	E3: 0.57 (−0.42–1.00)	E3: 83.3
8th left paravertebral line	E1: 0.75 (0.04–1.00)	E1: 91.7	E1: 0.57 (−0.42–1.00)	E1: 83.3
	E2: 1.00 (1.00–1.00)	E2: 100	E2: 1.00 (1.00–1.00)	E2: 100
	E3: 0.75 (0.04–1.00)	E3: 91.7	E3: 0.57 (−0.42–1.00)	E3: 83.3
8th right subscapular line	E1: 0.78 (0.20–1.00)	E1: 91.7	E1: 0.67 (−0.14–1.00)	E1: 83.3
	E2: 0.71 (0.18–1.00)	E2: 88.9	E2: 0.67 (−0.14–1.00)	E2: 83.3
	E3: 0.75 (0.04–1.00)	E3: 91.7	E3: 0.57 (−0.42–1.00)	E3: 83.3
8th left subscapular line	E1: 1.00 (1.00–1.00)	E1: 100	E1: 1.00 (1.00–1.00)	E1: 100
	E2: 1.00 (1.00–1.00)	E2: 100	E2: 1.00 (1.00–1.00)	E2: 100
	E3: 0.75 (0.04–1.00)	E3: 91.7	E3: 0.57 (−0.42–1.00)	E3: 83.3
8th right posterior axillary line	E1: 0.78 (0.20–1.00)	E1: 91.7	E1: 0.67 (−0.14–1.00)	E1: 83.3
	E2: 1.00 (1.00–1.00)	E2: 100	E2: 1.00 (1.00–1.00)	E2: 100
	E3: 0.78 (0.21–1.00)	E3: 91.7	E3: 0.67 (−0.14–1.00)	E3: 83.3
8th left posterior axillary line	E1: 1.00 (1.00–1.00)	E1: 100	E1: 1.00 (1.00–1.00)	E1: 100
	E2: 0.71 (0.18–1.00)	E2: 88.9	E2: 0.67 (−0.14–1.00)	E2: 83.3
	E3: 1.00 (1.00–1.00)	E3: 100	E3: 1.00 (1.00–1.00)	E3: 100
**Semiquantitative and Binary Pleural Line Scores, Respectively**	E1: 0.91 (0.27–1.00)	E1: 96.7	E1: 0.84 (0.56–1.00)	E1: 93.3
	E2: 0.86 (0.23–1.00)	E2: 94.4	E2: 0.77 (0.44–1.00)	E2: 90.0
	E3: 0.88 (0.24–1.00)	E3: 95.4	E3: 0.77 (0.44–1.00)	E3: 90.0

CI: confidence interval; E1: explorer 1; E2: explorer 2; E3: explorer 3.

**Table 4 jcm-14-03701-t004:** Inter-explorer reliability of the pleural line scores.

Pleural Line Evaluations	Semiquantitative Score		Binary Score	
	Kappa (95%CI)	Agreement %	Kappa (95%CI)	Agreement %
**Intercostal Spaces**				
**Anterolateral Thorax**				
2nd right parasternal line	0.73 (0.48–1.00)	92.1	0.70 (0.35–1.00)	85.7
2nd left parasternal line	0.89 (0.72–1.00)	96.8	0.78 (0.47–1.00)	90.5
4th right mid-clavicular line	0.81 (0.57–1.00)	95.2	0.87 (0.58–1.00)	95.2
4th left mid-clavicular line	1.00 (1.00–1.00)	100	1.00 (1.00–1.00)	100
4th right anterior axillary line	0.78 (0.56–1.00)	93.7	0.89 (0.64–1.00)	95.2
4th left anterior axillary line	0.71 (0.39–1.00)	92.1	0.75 (0.37–1.00)	90.5
4th right mid-axillary line	0.77 (0.54–1.00)	93.7	0.78 (0.47–1.00)	90.5
4th left mid-axillary line	0.77 (0.51–1.00)	92.9	0.77 (0.43–1.00)	90.5
**Posterior Thorax**				
8th right paravertebral line	0.82 (0.56–1.00)	95.2	0.66 (0.28–1.00)	85.7
8th left paravertebral line	0.94 (0.79–1.00)	98.4	0.88 (0.62–1.00)	95.2
8th right subscapular line	0.89 (0.70–1.00)	96.8	0.80 (0.51–1.00)	90.5
8th left subscapular line	0.80 (0.57–1.00)	93.6	0.62 (0.26–1.00)	80.9
8th right posterior axillary line	0.92 (0.78–1.00)	96.8	0.81 (0.53–1.00)	90.5
8th left posterior axillary line	0.96 (0.90–1.00)	98.4	0.90 (0.69–1.00)	95.2
**Semiquantitative and Binary Pleural Scores, Respectively**	0.84 (0.71–1.00)	94.2	0.84 (0.69–1.00)	93.3

CI: confidence interval.

**Table 5 jcm-14-03701-t005:** Diagnostic performance of the proposed LUS scores.

LUS Score	Cut-Off	AUROC	Sensitivity	Specificity
B-Line Global Score	≥3	0.74–0.77	60–80%	75%
B-Line Binary Score	≥1	0.80	60%	100%
Semiquantitative Pleural Line Score	≥2	0.82–0.87	90%	50–75%
Binary Pleural Line Score	≥4	0.77–0.81	60–80%	75%

AUROC: area under the receiver operating characteristic curve; LUS: lung ultrasound. In the cases where results differed between explorers, the maximum and minimum values obtained are indicated.

**Table 6 jcm-14-03701-t006:** Main characteristics of the studies on the reliability of lung ultrasound for RA-ILD screening in rheumatoid arthritis.

Authors, Date	Patients Included	US Machines, Scanning Protocol and Scores	Reliability
Cogliati et al., 2014 [22]	RA patients with suspected or known ILD	(1) Standard US machine and convex probe - 72 ICSs: BL total sum; if ≥5 BLs in a segment or confluent BL, value = 10 BLs; Total BL score > 10: positive exam (presence of ILD) - 8 ICSs (4 areas on each side of the anterolateral chest): ≥3 BLs in the same scan = positive area; ≥2 positive regions bilaterally = positive exam (ILD presence) (2) Pocket-sized US machine and phased array probe 72 ICS: A total BL score > 10 = positive examination (presence of ILD)	Prior to the study, inter-observer variability r = 0.96 for BL quantification (72 ICS) Inter-explorer kappa = 0.78 for the presence of ILD (expert physicians with standard US equipment versus briefly trained physicians with a pocket-sized US machine)
Moazedi-Fuerst et al., 2014 [23]	RA patients without clinical or radiographic suspicion of ILD and HC	Standard US machine, convex probe for parenchyma and linear probe for pleural line, 18 ICS Pathology definitions: BL in >2 locations, pleural line thickening (>2.8 mm), pleural line fragmentation, subpleural nodules and negative lung sliding	Inter-observer kappa = 0.92 for the absence/presence of ILD
Gutiérrez et al., 2022 [29]	RA patients without a previous history of acute or chronic pulmonary diseases and HC	Standard US machine, 14 ICSs, linear probe Semiquantitative BL score: 0 = normal (≤5 BL), 1 = slight (≥6 and ≤15 BL), 2 = moderate (≤16 and ≥30 BL), and 3 = severe (≥30 BL)	Prior to the study, 8 patients with different CTD-ILD, with inter-observer kappa = 0.82
Bandinelli et al., 2024 [40]	RA patients with mild respiratory symptoms	Standard US machine, 14 ICSs, linear probe and semiquantitative pleural (PLUS) and parenchymal (PAUS) scores #	US examiner: intra-reader ICC > 0.9 for both PLUS and PAUS; Trained residents: inter-reader ICC of 0.82–0.84 for PLUS and 0.86–0.94 for PAUS
Zabaleta et al., 2024 [41]	RA patients with chest HRCT in the 12 months prior to inclusion, regardless of symptomatology	Standard US machine, 14 ICSs, linear and convex probes Scores: Total number of BLs and pleural irregularities (PIs)	Prior to the study, inter-observer ICC = 0.97 for BL and ICC = 0.78 for PI and intra-observer ICC = 0.76 for BL ICC = 0.79 for PI
Watanabe et al., 2025 [42]	CTD-ILD patients with chest HRCT and LUS within an interval of <3 months	Standard US machine, 14 ICSs and microconvex probe Score: total number of BLs	Inter-rater ICC = 0.93 for the total B-line measurements

BLs: B lines; CTD: connective tissue disease; HC: healthy control; HRCT: high-resolution computerized tomography; ICC: intraclass correlation coefficient; ICSs: intercostal spaces; ILD: interstitial lung disease; LUS: lung ultrasound; RA: rheumatoid arthritis: US: ultrasound. # PLUS score: 1, non-linear, non-homogeneous and thickened pleural line; 2, disrupted pleural line (“fragmented”); and 3, subpleural consolidation (subpleural echo-poor region or “tissue-like”). PAUS score: 1, discrete divergent B lines; 2, confluent B lines; and 3, dense confluent areas (“whiteout”) that persist during the respiratory cycle.

## Data Availability

The datasets used and/or analyzed during the current study are available from the corresponding authors upon reasonable request.

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
