# Peer review of "A Proposal for a New Lung Ultrasound Score in Rheumatoid Arthritis: The Reliability of Lung Ultrasound for Rheumatoid Arthritis-Associated Interstitial Lung Disease Diagnosis"

_jcm, 2025, doi:10.3390/jcm14113701_

Round 1
Reviewer 1 Report
Comments and Suggestions for Authors
I appreciated having the opportunity to review this study that evaluates the intra- and inter-explorer reliability of several lung ultrasound scoring methods in patients with rheumatoid arthritis-associated interstitial lung disease.
I just have some minor suggestions for the authors that I think can help improve the manuscript overall.
-While US is a great, risk free technique, the authors should briefly acknowledge the limitations of lung ultra sound, especially in detecting fibrotic patterns compared to HRCT.
-The authors should clarify how blinding was enforced between examinations
- The authors should better explain the decision to define ≥3 B-lines as a pathological finding. Though it sounds reasonable and appears to follow expert consensus, however, this rationale would benefit from being explicitly linked to published guidelines or widely recognized criteria.
-Finally, the exact timeframe of recruitment and potential selection biases should be clarified by the authors of the manuscript
Author Response
RESPONSE TO REVIEWER 1
Comments and Suggestions for Authors
I appreciated having the opportunity to review this study that evaluates the intra- and inter-explorer reliability of several lung ultrasound scoring methods in patients with rheumatoid arthritis-associated interstitial lung disease.
We thank the reviewer for his/her interest in reviewing our work and his/her comments for improvement.
I just have some minor suggestions for the authors that I think can help improve the manuscript overall.
-While US is a great, risk free technique, the authors should briefly acknowledge the limitations of lung ultra sound, especially in detecting fibrotic patterns compared to HRCT.
We thank the reviewer for his/her comment. We agree with the reviewer that HRCT is the gold standard technique to detect lung fibrotic patterns, and that LUS has limitations compared to HRCT. We have included the following sentences about the limitations of LUS:
- Page 19 (Lines 548–556) of revised version with highlighted changes:
“Finally, the inherent limitations of LUS in its ability to detect pulmonary fibrotic patterns compared to HRCT cannot be completely ruled out. B lines may be present in other pathologies such as heart failure, pulmonary infections, pleural disease, or atelectasis, which limits its specificity, so a thorough clinical evaluation is necessary to exclude these processes. Furthermore, in healthy elderly patients, a limited number of B lines can also be found, which can complicate the interpretation of LUS in patients with suspected RA-ILD due to its association with age, as one of its risk factors. It should be added that LUS, unlike HRCT, cannot evaluate the entire parenchyma and has limitations for differential diagnosis with infection and cancer”.
-The authors should clarify how blinding was enforced between examinations
We thank the reviewer for pointing out this crucial aspect related to the blinding of ultrasound exploration. We have clarified it by adding the following explanation:
- Page 3 (Lines 122–125) of revised version with highlighted changes:
“To reinforce blinding between explorers, neither was present during the other's examinations. Furthermore, to reinforce blinding of the sonographers for the intra-explorer study, the examination room was darkened, and patients were instructed not to exchange comments with the sonographers.”
- The authors should better explain the decision to define ≥3 B-lines as a pathological finding. Though it sounds reasonable and appears to follow expert consensus, however, this rationale would benefit from being explicitly linked to published guidelines or widely recognized criteria.
We thank the reviewer for identifying an area that can benefit from more detailed information. Probably the fact that we mentioned reference 38 by the committee created to generate the document – International Liaison Committee on Lung Ultrasound (ILC-LUS) for the International Consensus Conference on Lung Ultrasound (ICC-LUS) – may have created confusion. This reference refers to the International Evidence-Based Recommendations for Point-of-Care Lung Ultrasound. Statements were discussed and elaborated by experts who published most papers on clinical use of lung ultrasound in the previous 20 years. Recommendations were produced to guide implementation, development, and standardization of lung ultrasound in all relevant settings. Regarding LUS evaluation of B lines for interstitial syndrome, a positive region is defined by the presence of three or more B-lines in a longitudinal plane between two ribs.
At this time, when a consensus has not yet been reached on the optimal LUS method for diagnosing RA-ILD, we found it interesting to test a widely used approach for diagnosing interstitial lung syndrome in other medical settings, such as emergency department assessment and assessment of critically ill patients.
We have included the following explanations and references:
- Page 5 (Lines 176–182) of revised version with highlighted changes:
“following the definition of a positive region established by the International Evidence-Based Recommendations for Point-of-Care Lung Ultrasound, agreed upon by experts and widely used in other medical fields such as the assessment of interstitial syndrome in emergencies and in critically ill patients [38].”
- Page 16 (Line 391) of revised version with highlighted changes:
“decided the scanning protocol and generated the corresponding scores, relying on the international evidence-based recommendations for point-of-care LUS [38], …”
-Finally, the exact timeframe of recruitment and potential selection biases should be clarified by the authors of the manuscript
We thank the reviewer for his/her comment. We have included the requested information in the following lines of the revised manuscript:
- Page 3 (Line 113) of revised version with highlighted changes:
“The recruitment period was from May to July 2022.”
- Page 19 (Lines 545–548) of revised version with highlighted changes:
“Due to the recruitment of patients being consecutive and without randomization, the existence of a selection bias cannot be ruled out. Additionally, it did not allow for stratification by RA-ILD severity, although its impact on reliability might not be possible to delineate with the limited sample size.”

Reviewer 2 Report
Comments and Suggestions for Authors
This is an interesting original research article investigating the implementation of lung ultrasound (LUS) in the diagnostic workup of rheumatoid arthritis (RA) patients suspected for RA-associated interstitial lung disease (RA-ILD). More specifically, the authors investigated the reliability of new proposed LUS scores based on the assessment of B-lines and the pleural line in suspected cases of RA-ILD. The description of the background knowledge, explanation of the research question, methodology of LUS performance, and statistical analysis are well presented in the manuscript. However, there are major points that the authors are advised to reconsider:
1) It is inaccurate to refer to LUS as a screening tool in this study, as the participants were not asymptomatic RA patients but rather RA patients with suspected RA-ILD based on their signs, symptoms, and pulmonary function testing, as noted in the manuscript. It is better to refer to LUS as a diagnostic tool in this case.
2) It is important to more clearly specify the clinical settings of LUS performance, e.g. if it was done on an inpatient or outpatient basis and if a recent respiratory tract infection or other acute pulmonary event potentially affecting the findings of LUS was considered as an exclusion criterion.
3) Reporting the presence or absence of RA-ILD based on the gold standard diagnostic test, i.e. chest HRCT, is necessary. Although this study investigates the precision, i.e. reliability, rather than the accuracy, i.e. validity, of LUS for RA-ILD, it is still important to clarify whether the participants were finally diagnosed with RA-ILD, as the reliability of the test could differ between RA patients with or without associated ILD. Further, the reliability of LUS could also be affected by the different levels of severity of RA-ILD, although this might not be possible to delineate in this small sample size.
4) The systematic review incorporated in the study adds a significant value to the overall article, taking into consideration the limitations of the study, especially the small sample size. For this reason, this part of the study should better be improved by registering the systematic review in an official database, e.g. PROSPERO, including at least two databases in addition to PubMed, i.e. at least three databases in total, and explaining in more detail the screening process of the records and the data extracted from the included studies [Page, M.J., McKenzie, J.E., Bossuyt, P.M., et al. (2021). The PRISMA 2020 statement: an updated guideline for reporting systematic reviews. BMJ, 372.]
5) Future perspectives of the investigation of LUS reliability in the diagnosis of RA-ILD could be reported in the last part of the Discussion, e.g. the validation of the results in larger sample sizes and the comparison of LUS reliability between different levels of severity of RA-ILD or between RA-ILD and other systemic autoimmune rheumatic disease-associated -ILD.
5) Minor comment: Although the manuscript is generally well-written, it should be screened for minor linguistic errors. The abbreviated terms “RA” and “ILD” should better be explained in the title as well.
Author Response
RESPONSE TO REVIEWER 2
Comments and Suggestions for Authors
This is an interesting original research article investigating the implementation of lung ultrasound (LUS) in the diagnostic workup of rheumatoid arthritis (RA) patients suspected for RA-associated interstitial lung disease (RA-ILD). More specifically, the authors investigated the reliability of new proposed LUS scores based on the assessment of B-lines and the pleural line in suspected cases of RA-ILD. The description of the background knowledge, explanation of the research question, methodology of LUS performance, and statistical analysis are well presented in the manuscript.
We thank the reviewer for these favorable comments about our manuscript.
However, there are major points that the authors are advised to reconsider:
1) It is inaccurate to refer to LUS as a screening tool in this study, as the participants were not asymptomatic RA patients but rather RA patients with suspected RA-ILD based on their signs, symptoms, and pulmonary function testing, as noted in the manuscript. It is better to refer to LUS as a diagnostic tool in this case.
We thank the reviewer for his/her comment aiming to improve our manuscript.
The reviewer is right in his/her appraisal. These patients were tested to determine whether their signs or symptoms were due to the development of RA-associated ILD. We have changed it in the following paragraphs:
- Page 1 (Lines 2–5) of revised version with highlighted changes: we have changed the title to “Proposal for a new score of the lung ultrasound in Rheumatoid Arthritis. Reliability of Lung Ultrasound for Rheumatoid Arthritis-associated Interstitial Lung Disease diagnosis”
- Page 2 (Lines 48–50) of revised version with highlighted changes: we have changed the conclusions of the Abstract.
“Conclusions: Substantial to excellent intra- and inter-explorer reliability of different feasible B-line and pleural LUS scores were found, adding evidence in favor of the potential implementation of LUS for RA-ILD diagnosis in clinical practice.”
- Page 2 (Lines 83–85) of revised version with highlighted changes:
“In line with literature, a previous research study from our group carried out in 192 patients with RA and suspected RA-ILD also suggested the utility of LUS for ILD diagnosis ……………….”.
- Page 3 (Line 93) of revised version with highlighted changes:
“Regarding the reliability of LUS for RA-ILD screening or diagnosis, ………….”.
- Page 5 (Lines 196–197) of revised version with highlighted changes:
“The systematic literature review searched for original papers about the use of LUS for RA-ILD screening or diagnosis published in PUBMED ……………”.
- And throughout the manuscript.
2) It is important to more clearly specify the clinical settings of LUS performance, e.g. if it was done on an inpatient or outpatient basis and if a recent respiratory tract infection or other acute pulmonary event potentially affecting the findings of LUS was considered as an exclusion criterion.
We thank the reviewer for identifying an area that can benefit from more detailed information. The study was performed on an outpatient basis, and a recent respiratory tract infection or other acute pulmonary events potentially affecting the findings of LUS were considered as an exclusion criterion.
We have included the following explanations:
- Page 3 (Lines 105–113) of revised version with highlighted changes:
“All patients met the 2010 ACR/ European Alliance of Associations for Rheumatology (EULAR) classification criteria [30] and were undergoing routine clinical monitoring of their disease on an outpatient basis at the IPS Biomab. RA-ILD suspicion was based on the presence of velcro crackles or respiratory symptoms (chronic cough and/or dyspnea) in the last two months that could not be explained by a recent respiratory process (heart failure, infection, pleural effusion, atelectasis, asthma or chronic obstructive pulmonary disease), or LFT or imaging alterations. Pregnancy and previous interstitial lung involvement were additional exclusion criteria.”
- Page 3 (Lines 113–114):
“Eligible patients who agreed to participate in the study underwent analysis, LFT, HRCT and LUS.”.
3) Reporting the presence or absence of RA-ILD based on the gold standard diagnostic test, i.e. chest HRCT, is necessary. Although this study investigates the precision, i.e. reliability, rather than the accuracy, i.e. validity, of LUS for RA-ILD, it is still important to clarify whether the participants were finally diagnosed with RA-ILD, as the reliability of the test could differ between RA patients with or without associated ILD. Further, the reliability of LUS could also be affected by the different levels of severity of RA-ILD, although this might not be possible to delineate in this small sample size.
We thank the reviewer for the suggestion of giving information about chest HRCT that was performed in all patients.
We did not include this information in the paper because, as the reviewer points out, our work focused on studying the precision of lung ultrasound and not its accuracy. However, we appreciate the suggestion to improve our manuscript and have provided information regarding the presence or absence of ILD on HRCT. As the reviewer points out, the small sample size of our pilot study does not allow us to delineate the impact of the different degrees of severity of RA-ILD.
Given that our patients underwent all the standard tests for ILD diagnosis, including HRCT, we performed this analysis in our patients. We have calculated the area under the ROC curve for the different scores evaluated, identifying the cutoff points that allowed the discrimination of the presence of ILD on HRCT. Despite the inherent limitations of this being a pilot study focused on analyzing the intra- and inter-examiner reliability of lung ultrasound, the values obtained were generally good for all scores and examiners, ranging from 0.74 to 0.87. We have included an additional table (Table 5) with the cut-off points, sensitivity, specificity and AUC, and a comment on the interest in confirming these data with a study with a larger sample size that includes patients stratified for different degrees of lung involvement, since our sample size does not allow us to perform such an analysis. We have also included this assessment in the statistical analysis section.
- Page 6 (Lines 227–230) of revised version with highlighted changes:
“The ability of our LUS scores to discriminate the presence of ILD in HCRT was assessed using receiver operating characteristic (ROC) curves, and optimal cut-off values were determined. The sensitivity, specificity, and area under the ROC curve (AUROC) were considered as diagnostic performance indicators.”
- Page 11 (Lines 284–292) of revised version with highlighted changes:
“ 3.3. Diagnostic performance of the proposed LUS scores
Although the objective of our research focused on reliability and not the accuracy of LUS, we evaluated whether our proposed LUS scores could discriminate against the presence of ILD on HRCT, considered the gold standard for the diagnosis of RA-ILD. HRCT was normal in 4 patients, showing ILD in the rest. The values of the AUROC curve were generally good for all scores and examiners, ranging from 0.74 to 0.87. Table 5 shows the optimal cutoff points for each LUS score, obtained using ROC curves, to discriminate between patients with and without ILD on the HRCT, the AUROC, the sensitivity, and specificity.”
- Pages 11-12 (Lines 293–296) of revised version with highlighted changes:
Table 5. Diagnostic performance of the proposed LUS scores.
|
LUS score |
Cut-off |
AUROC |
Sensitivity |
Specificity |
|
B LINES GLOBAL SCORE |
≥ 3 |
0.74 - 0.77 |
60% - 80% |
75% |
|
B LINES BINARY SCORE |
≥ 1 |
0.80 |
60% |
100% |
|
SEMIQUANTITATIVE PLEURAL LINE SCORE |
≥ 2 |
0.82 - 0.87 |
90% |
50% - 75% |
|
BINARY PLEURAL LINE SCORE |
≥ 4 |
0.77 - 0.81 |
60% - 80% |
75% |
AUROC: area under the receiver operating characteristic curve; LUS: lung ultrasound. In cases where results differed between explorers, the maximum and minimum values ​​obtained have been indicated.
4) The systematic review incorporated in the study adds a significant value to the overall article, taking into consideration the limitations of the study, especially the small sample size. For this reason, this part of the study should better be improved by registering the systematic review in an official database, e.g. PROSPERO, including at least two databases in addition to PubMed, i.e. at least three databases in total, and explaining in more detail the screening process of the records and the data extracted from the included studies [Page, M.J., McKenzie, J.E., Bossuyt, P.M., et al. (2021). The PRISMA 2020 statement: an updated guideline for reporting systematic reviews. BMJ, 372.]
We thank the reviewer for helping us improve our manuscript.
We have created an account in PROSPERO to register our systematic review. Unfortunately, we are unable to register our review at this time, as the instructions clearly state that reviews must be registered before data extraction has started or before IPD has been received. It also states that action will be taken if inaccuracies in the data are identified at any time, including information supplied about the stage of review or anticipated completion time. We appreciate the information provided by the reviewer regarding the PROSPERO resource, which we will use in planning future systematic reviews and/or meta-analyses.
We have done the search in 3 databases: PUBMED, EMBASE and Cochrane as reflected in the revised Figure 2 about the systematic review flow chart. No additional papers have been identified in the added databases. We have also included the search strategy in both additional databases, and explained in more detail the screening process of the records and the data extracted from the included studies:
The changes can be seen in those paragraphs:
- Pages 5-6 (Lines 196–221) of revised version with highlighted changes:
“The objective of the systematic review was to gather the existing evidence in literature about reliability of LUS applied to screening or diagnosis of RA-ILD. We searched for original papers about the use of LUS for RA-ILD screening or diagnosis published in PUBMED, EMBASE or Cochrane Library until the end of December 2024, using the following search strategies according to each database: a) PUBMED: (“rheumatoid arthritis”[Title/Abstract] AND (“interstitial lung disease”[Title/Abstract] OR “ILD”[Title/Abstract] OR “usual interstitial pneumo*”[Title/Abstract]) AND (“ultrasound”[Title/Abstract] OR “ultrasonography*”[Title/Abstract])) AND (1000/1/1:2024/12/31[pdat]); b) EMBASE: 'rheumatoid arthritis':ab,ti AND ('interstitial lung disease':ab,ti OR 'ild':ab,ti OR 'usual interstitial pneumonia':ab,ti) AND ('ultrasound':ab,ti OR 'ultrasonography':ab,ti) AND ('article'/it OR 'article in press'/it OR 'clinical trial'/it OR 'erratum'/it OR 'review'/it) AND [<1966-2024]/py, excluding conference abstracts and conference reviews; and c) Cochrane Library: “rheumatoid arthritis” AND (“interstitial lung disease” OR “ILD” OR (usual interstitial NEXT pneumo*)) AND (“ultrasound” OR “ultrasonography”), in Title Abstract Keyword and restricted by publication date from January 1966 to December 2024.
The article selection process began with a review of the title and abstract. If information on the reliability of LUS was not included, the materials and methods sections, and finally the results, were reviewed. Only original manuscripts in which the reliability of LUS for RA-ILD screening or diagnosis had been measured and reported were selected, excluding reviews, meta-analysis, editorials and case reports. Despite the reviews not being eligible for inclusion, they were revised for potential references of interest that met the inclusion criteria and had not been identified by the search strategy. The following data were extracted from the selected articles: author, year of publication, patients included, type of ultrasound machines used, scanning protocols and scores, and reliability data.”
- Page 12 (Lines 304–307) of revised version with highlighted changes:
“3.4. Systematic literature review
Forty manuscripts were identified by the search strategy in PUBMED and EMBASE databases and 2 manuscripts in Cochrane Library. Six eligible papers were selected after excluding those that did not meet the inclusion criteria and duplicates (Figure 2).”
- Page 12 (Lines 313–314) of revised version with highlighted changes:
No additional original articles were identified after revising the reviews.
- Page 13 (Lines 315) of revised version with highlighted changes:
Previous Figure 2 has been replaced by the revised Figure 2.
Figure 2. Systematic review flow chart. LUS: lung ultrasound. RA-ILD: rheumatoid arthritis with associated interstitial lung disease.
5) Future perspectives of the investigation of LUS reliability in the diagnosis of RA-ILD could be reported in the last part of the Discussion, e.g. the validation of the results in larger sample sizes and the comparison of LUS reliability between different levels of severity of RA-ILD or between RA-ILD and other systemic autoimmune rheumatic disease-associated -ILD.
We thank the reviewer for the interesting suggestion. We have included a comment about future perspectives in our manuscript:
- Pages 18-19 (Line 522–527) of revised version with changes highlighted:
“As a future perspective, given the favorable findings of our pilot study, we believe it would be of great interest to conduct further research to validate these results in studies with larger sample sizes, comparing LUS reliability between different levels of severity of RA-ILD, including patients in the preclinical or early stages of the disease. It would also be interesting to compare reliability between patients with RA-ILD and other systemic autoimmune rheumatic disease-associated ILD.”
5) Minor comment: Although the manuscript is generally well-written, it should be screened for minor linguistic errors. The abbreviated terms “RA” and “ILD” should better be explained in the title as well.
We are grateful to the reviewer for identifying these minor linguistic mistakes which we have corrected:
- Page 1 (Line 2–5) of revised version with highlighted changes: We have changed the title to “Proposal for a new score of the lung ultrasound in Rheumatoid Arthritis. Reliability of Lung Ultrasound for Rheumatoid Arthritis-associated Interstitial Lung Disease diagnosis”.
- We have made minor corrections throughout the manuscript.

Reviewer 3 Report
Comments and Suggestions for Authors
Dear authors,
You provide a manuscript that adresses a relevant topic, screening for ILD i rheumatoid arthritis.
However, significant limitations exist here and they are extremely small sample size (n=14), absence of correlation with HRCT findings (which are gold standard as you say). Clinical utility is small without comparison to HRCT.
Small sample size greatly limit generalizability of your conclusions. Should state that clearly in limitations. Altough reliability studies can be conducted on smaller samples, this limitation is important when results are used to imply potential clinical implementation. It would also help if patients were stratified for different degrees of lung involvement. If we use LUS for screening in RA we will that it can detect the early changes. Therefore it would be better to test also on many patients with small lung involvement to see how sensitive LUS can be.
The strength is that ultrasound protocol is sound and very well standardized. Excellent kappa values.
Author Response
RESPONSE TO REVIEWER 3
Comments and Suggestions for Authors
Dear authors,
You provide a manuscript that adresses a relevant topic, screening for ILD i rheumatoid arthritis.
However, significant limitations exist here and they are extremely small sample size (n=14), absence of correlation with HRCT findings (which are gold standard as you say). Clinical utility is small without comparison to HRCT.
We thank the reviewer for his/her comment. We agree with the reviewer that the sample size is small to investigate LUS accuracy. However, our pilot study has focused on intra- and inter-explorer reliability, as a crucial aspect to advance its validation process, since we have identified that this area is still in need of more scientific evidence. We would like to highlight that the complexity of these studies does not allow very big sample sizes.
In response to the reviewer's comment, we have clearly specified the limitation of the small sample size of our reliability pilot study in the limitations section:
- Page 19 (Lines 531–538) of revised version with highlighted changes:
“In our pilot reliability study, we applied measures at our disposal to mitigate the limitations of the small sample size, such as being conducted by experienced sonographers, and the previously mentioned measures used to increase its robustness. Organizing re-liability studies like ours, in which patients are examined twice by three explorers, is complex. They need to be conducted at a single site with a limited sample size and sufficient time (> 5 hours) to ensure blinding. However, the findings of our pilot study must be interpreted with caution and considered a starting point worthy of further investigation because its small sample size limits the generalizability of the conclusions.”
Regarding the comparison of LUS with HRCT, we did not include this information in the paper because our work focused on studying the reliabilty of lung ultrasound and not its accuracy. However, we agree with the reviewer that the comparison of LUS with HRCT is of greatest clinical interest. Given that our patients underwent all the standard tests for ILD diagnosis, including HRCT, we performed this analysis in our patients.
We have calculated the area under the ROC curve for the different scores evaluated, identifying the cutoff points that allowed the discrimination of the presence of ILD on HRCT. Despite the inherent limitations of this being a pilot study focused on analyzing the intra- and inter-examiner reliability of lung ultrasound, the values ​​obtained were generally good for all scores and explorers, ranging from 0.74 to 0.87. We have included an additional table (Table 5) with the cut-off points, sensitivity, specificity and AUC, and a comment on the interest in confirming these data with a study with a larger sample size that includes patients stratified for different degrees of lung involvement, since our sample size does not allow us to perform such an analysis. We have also included this assessment in the statistical analysis section.
- Page 6 (Lines 227–230) of revised version with highlighted changes:
“The ability of our LUS scores to discriminate the presence of ILD in HCRT was assessed using receiver operating characteristic (ROC) curves, and optimal cut-off values were determined. The sensitivity, specificity, and area under the ROC curve (AUROC) were considered as diagnostic performance indicators.”
- Page 11 (Lines 284–292) of revised version with highlighted changes:
“ 3.3. Diagnostic performance of the proposed LUS scores
Although the objective of our research focused on reliability and not the accuracy of LUS, we evaluated whether our proposed LUS scores could discriminate against the presence of ILD on HRCT, considered the gold standard for the diagnosis of RA-ILD. HRCT was normal in 4 patients, showing ILD in the rest. The values of the AUROC curve were generally good for all scores and examiners, ranging from 0.74 to 0.87. Table 5 shows the optimal cutoff points for each LUS score, obtained using ROC curves, to discriminate between patients with and without ILD on the HRCT, the AUROC, the sensitivity, and specificity.”
- Pages 11-12 (Lines 293–296) of revised version with highlighted changes:
Table 5. Diagnostic performance of the proposed LUS scores.
|
LUS score |
Cut-off |
AUROC |
Sensitivity |
Specificity |
|
B LINES GLOBAL SCORE |
≥ 3 |
0.74 – 0.77 |
60% – 80% |
75% |
|
B LINES BINARY SCORE |
≥ 1 |
0.80 |
60% |
100% |
|
SEMIQUANTITATIVE PLEURAL LINE SCORE |
≥ 2 |
0.82 – 0.87 |
90% |
50% – 75% |
|
BINARY PLEURAL LINE SCORE |
≥ 4 |
0.77 – 0.81 |
60% - 80% |
75% |
AUROC: area under the receiver operating characteristic curve; LUS: lung ultrasound. In cases where results differed between explorers, the maximum and minimum values ​​obtained have been indicated.
Small sample size greatly limit generalizability of your conclusions. Should state that clearly in limitations. Altough reliability studies can be conducted on smaller samples, this limitation is important when results are used to imply potential clinical implementation. It would also help if patients were stratified for different degrees of lung involvement. If we use LUS for screening in RA we will that it can detect the early changes. Therefore it would be better to test also on many patients with small lung involvement to see how sensitive LUS can be.
We thank the reviewer for his/her comment aimed at improving our manuscript. We have included the following paragraphs to state clearly in the limitations that the small sample size limits the generalizability of your conclusions:
- Page 19 (Lines 536–538) of revised version with highlighted changes:
“However, the findings of our pilot study must be interpreted with caution and considered a starting point worthy of further investigation because its small sample size limits the generalizability of the conclusions.”
We thank the reviewer for his/her suggestion about the interest of stratifying for different degrees of lung involvement. We agree that it would be very interesting to confirm our preliminary results obtained in this pilot study with a larger sample size study that includes patients stratified for different degrees of lung involvement. We have also included this aspect in our limitations, and as a future perspective:
- Page 19 (Lines 545–548) of revised version with highlighted changes:
“Due to the recruitment of patients being consecutive and without randomization, the existence of a selection bias cannot be ruled out. Additionally, it did not allow for stratification by RA-ILD severity, although its impact on reliability might not be possible to delineate with the limited sample size.”
- Pages 18-19 (Lines 522–527) of revised version with highlighted changes:
“As a future perspective, given the favorable findings of our pilot study, we believe it would be of great interest to conduct further research to validate these results in studies with larger sample sizes, comparing LUS reliability between different levels of severity of RA-ILD, including patients in the preclinical or early stages of the disease. It would also be interesting to compare reliability between patients with RA-ILD and other systemic autoimmune rheumatic disease-associated ILD.”
The strength is that ultrasound protocol is sound and very well standardized. Excellent kappa values.
We thank the reviewer for this favorable comment and for the suggestions made to improve our manuscript.
